# Cell signaling model for arterial mechanobiology

**Linda Irons** *, **Jay D. Humphrey**

Department of Biomedical Engineering, Yale University, New Haven, Connecticut, United States of America

* linda.irons@yale.edu

**Data Availability Statement:** MATLAB files are available at: https://github.com/irons-l/arterialsignaling.

**Funding:** This work was supported, in part, by grants awarded to JDH from the US National Institutes of Health (NIH): R01 HL105297, P01

## Abstract

Arterial growth and remodeling at the tissue level is driven by mechanobiological processes at cellular and sub-cellular levels. Although it is widely accepted that cells seek to promote tissue homeostasis in response to biochemical and biomechanical cues—such as increased wall stress in hypertension—the ways by which these cues translate into tissue maintenance, adaptation, or maladaptation are far from understood. In this paper, we present a logic-based computational model for cell signaling within the arterial wall, aiming to predict changes in extracellular matrix turnover and cell phenotype in response to pressure-induced wall stress, flow-induced wall shear stress, and exogenous sources of angiotensin II, with particular interest in mouse models of hypertension. We simulate a number of experiments from the literature at both the cell and tissue level, involving single or combined inputs, and achieve high qualitative agreement in most cases. Additionally, we demonstrate the utility of this modeling approach for simulating alterations (in this case knockdowns) of individual nodes within the signaling network. Continued modeling of cellular signaling will enable improved mechanistic understanding of arterial growth and remodeling in health and disease, and will be crucial when considering potential pharmacological interventions.

## Author summary

Biological soft tissues are characterized by continuous production and removal of material, which endows them with a remarkable ability to adapt to changes in their biochemical and biomechanical environments. For arteries, mechanical stimuli result primarily from changes in blood pressure or flow, and biochemical changes are induced by multiple factors, including pharmacological intervention. In order to understand how arterial properties are maintained in health, or how they adapt or fail to adapt in disease, we must understand better how these diverse stimuli affect material turnover. Extracellular matrix is tightly regulated by mechano-sensing and mechano-regulation, and therefore cell signaling, thus we present a computational model of relevant signaling pathways within the vascular wall, with the aim of predicting changes in wall composition and function in response to three main inputs: pressure-induced wall stress, flow-induced wall shear stress, and exogenous angiotensin II. We obtain qualitative agreement with a range of

HL134605, U01 HL142518, and R01 HL146723. The funders had no role in study design, data collection and analysis, decision to publish, or preparation of the manuscript.

**Competing interests:** The authors have declared that no competing interests exist.

experimental studies from the literature, and provide illustrative examples demonstrating how such models can be used to further our understanding of arterial remodeling.

## Introduction

Central arteries actively maintain their geometry, composition, properties, and function over long periods under normal conditions. Moreover, they often adapt well to altered mechanical loading via the turnover of cells and extracellular matrix (ECM) in evolving configurations. Both of these observations are consistent with mechanical homeostasis, which exists across scales from sub-cellular to cellular to tissue levels [1]. Because of the complexity of such growth and remodeling (G&R) processes, computational models have proven useful in quantifying and comparing responses across both normal adaptations and disease conditions, often at the tissue-level [2–5]. Although tissue-level models increase our understanding of the time-course of certain homeostatic mechanisms, and their loss in cases of disease, and enable clinically relevant predictions, they are yet limited because of the lack of consideration of the underlying cell signaling pathways. There is, therefore, a pressing need for cell signaling models that affect responses at the tissue level.

Detailed kinetic models of cell signaling networks require a comprehensive understanding of both network structure and the underlying biochemistry. When known, a mathematical description can be formulated as a system of coupled differential equations, where processes such as phosphorylation and gene transcription are modeled by proposing appropriate functional relations [6, 7]. One of the primary challenges to such modeling, however, is parameterization. This is particularly challenging when there is crosstalk within the network, which makes isolating individual reactions and parameters difficult. These difficulties are manageable for smaller systems through parameter estimation and qualitative parameter explorations. For larger systems, or systems for which the precise nature of interactions are not well-understood, logic-based models represent an alternative approach that can offer significant insight [8–10]. These models also use a network structure, but are typically built using qualitative relationships between species, relying on more general observations such as 'A upregulates B' or 'C inhibits D', as often reported in the literature on vascular biology. Such models comprise a discrete set of rules together with an updating scheme for the state of each variable. Notably, precise functional forms and values of rate parameters for the interactions between species need not be known.

In this paper, we propose a logic-based model for the arterial wall focusing on signaling pathways that dominate responses to changes in mechanical loading at the tissue level as well as exposure to exogenous angiotensin II (AngII). In particular, chronic infusion of AngII is often used to induce hypertension in mouse models and we designed our computational model to consider simultaneously the potential roles of altered wall stresses, wall shear stresses, and AngII infusion on changes in intramural cell phenotype and turnover of ECM. Specifically, the network structure that defines the model is motivated by cell-level findings reported in 72 complementary studies in the literature, then values of the model parameters are tuned based on cell- and tissue-level findings from an additional, independent, set of 37 papers that report qualitative outcomes in terms of single perturbed inputs. Finally, we simulate case study experiments at both cell and tissue levels to enable qualitative validation studies using some of the most complete data available; these studies consider multiple perturbed inputs and multiple perturbation magnitudes.

## Results

### Network structure

First, consider a first generation network appropriate for studying arterial G&R (Fig 1), with input and output nodes relevant to arterial signaling in response to changes in mechanical loading and a possible exogenous source of AngII. It is well established that both intramural stress and wall shear stress (arising from blood pressure and flow, respectively) play important biomechanical roles in regulating wall geometry, composition, properties, and contractile function [11–13]. For example, intramural stress triggers medial smooth muscle cell (SMC) and adventitial fibroblast (FB) signaling pathways mediated by integrins and stretch-activated channels (SACs); it also affects the availability and activation of latent transforming growth factor-β (TGFβ), AngII, and platelet-derived growth factor (PDGF) [14–16], among other mediators. Wall shear stress is sensed primarily by endothelial cells (ECs); modestly decreased or increased shear stress results in the production of endothelin-1 (ET1) or nitric oxide (NO),

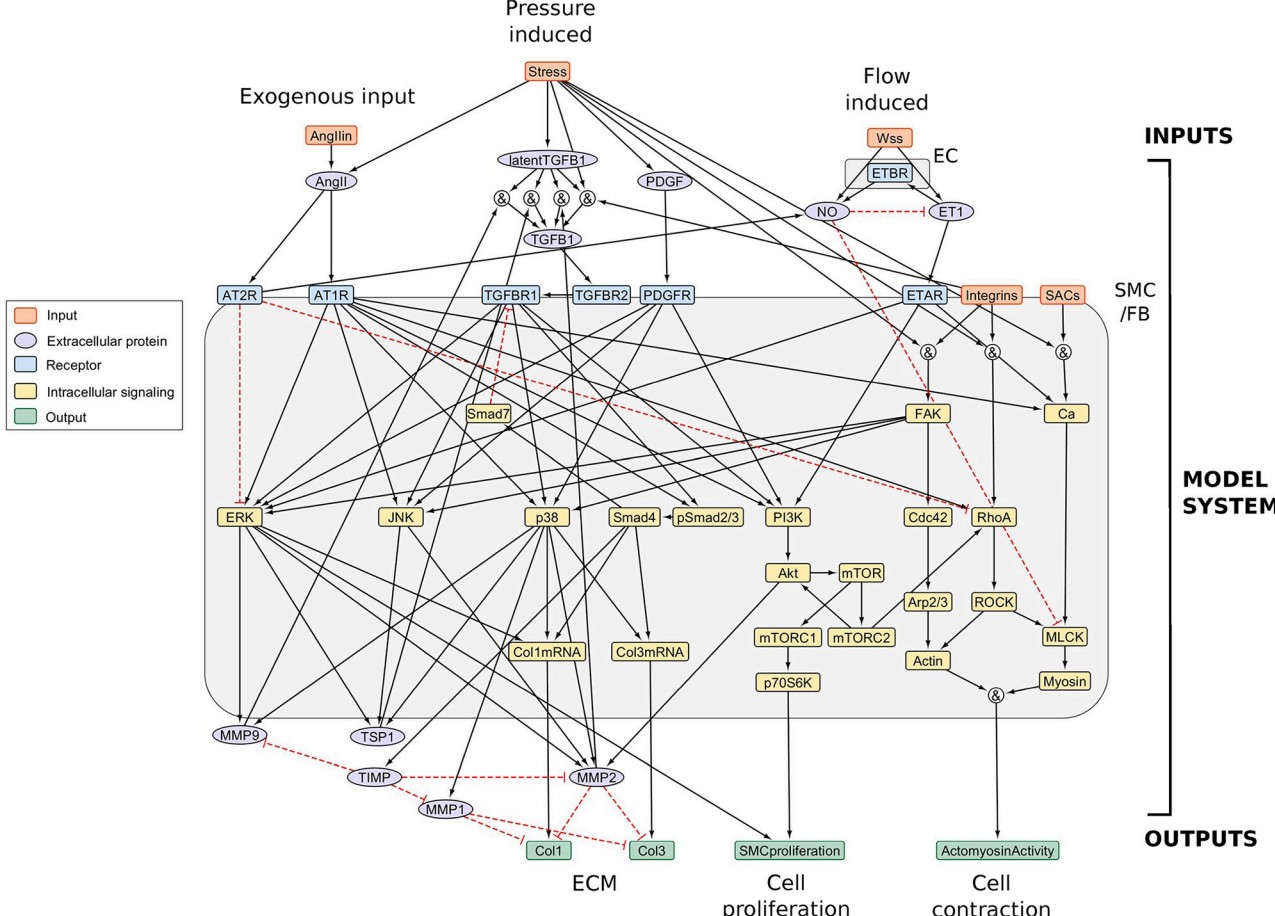

**Fig 1. Arterial wall signaling network constructed from the literature.** The network structure corresponds to rule-based statements, derived from the literature and shown (along with abbreviations) in S1 Appendix, containing 50 species and 82 reactions. Black solid lines denote activation and red dotted lines inhibition. For clarity, inhibition is shown to affect a node directly; however, to implement this, an 'AND NOT' logic operation is used with all incoming reactions to the node (see S1 Appendix). EC represents endothelial cells (the signaling for which is not considered in detail—see text) and SMC/FB refers to a homogenized approach to modeling contributions by the intramural smooth muscle cells and fibroblasts. Given our interest in AngII-induced hypertension, we focus on collagen production leading to fibrosis as revealed by murine experiments. Network visualization was achieved using Cytoscape [19] and Netflux (https://github.com/saucermanlab/Netflux).

a vasoconstrictor and vasodilator, respectively [17, 18]. Of particular relevance to murine models of hypertension, aneurysms, and dissection, we additionally consider an exogenous supply of AngII. In a commonly used murine model, AngII is infused chronically *in vivo* via an implanted mini-osmotic pump; our inclusion of AngII as an independent external source thus allows investigation of downstream signaling effects induced by different doses.

Based on these observations, we consider three main input species: intramural stress, wall shear stress, and exogenous AngII. For the intermediate species, with specific pathways discussed below, we consider EC responses to flow, and intramural cell (SMC/FB) responses to intramural stress and AngII. As illustrated in Fig 1, our EC component does not yet consider the cell signaling pathways in detail. Its purpose, rather, is to model the production of NO and ET1 that affect intramural cell signaling in general mechano-adaptations. Our simplifying choice to begin by considering the intramural cells together is motivated by prior mechanical models wherein mean (radially homogenized) wall mechanics are captured well using a single-layered model [20], and prior mechanobiological models for G&R [21–23] wherein salient features of arterial adaptations and maladaptations are captured using phenomenological models of combined SMC and FB mechanobiology, which reside in the medial and adventitial layers, respectively. This starting point is also convenient for future coupling of our signaling model to such G&R models, which we address in more detail in the Discussion.

The edges in the network (Fig 1) were constructed from the literature, and a full list of network relations and supporting references are provided in S1 Appendix. To formally define the model, relations were formulated as a set of 82 logic statements (see Methods). Many vascular diseases exhibit altered ECM and modified contractile function, thus our literature search focused on pathways relevant to collagen deposition and degradation (via matrix metalloproteinases) by SMCs/FBs as well as SMC contractility and proliferation, namely the Smad, MAPK (p38, ERK, JNK), PI3K, mTOR, and Rho/ROCK pathways. The majority of edges (62 of 82 relations) were deduced directly from experimental studies using vascular smooth muscle cells, vascular endothelial cells, or vascular tissue samples (see S1 Appendix). Where possible, we focused on murine studies. Some relations, such as the binding of a protein to its receptor, are well established (14 of 82 relations), and we cite relevant reviews for these more general phenomena. Additional relations were considered if there was evidence across multiple cell types and if these relations had previously been proposed to hold in reviews of vascular biology (5 of 82 relations). Finally, one edge (AngIIin $\Rightarrow$ AngII) was a model specified reaction (1 of 82 relations) to allow both exogenous (*e.g.* via a mini-osmotic pump) and endogenous sources. From this starting point, we envisage having to iteratively refine the network structure for additional situations of interest and as new data become available; the present rule-based approach provides a convenient means to do this.

### Input–output relations

Consider, first, how the network model can match experimental bio-chemo-mechanical input–output relations from the vascular literature. Most commonly in experiments, one variable is perturbed at a time from a baseline value, then comparisons are made amongst the outputs at baseline and following the perturbation, as, for example, an increase in stretch, flow, pressure, or an exogenous input such as AngII. In the model, we can also prescribe time-varying input values and measure corresponding changes in the outputs. To simulate such experiments, initial conditions for the inputs were prescribed such that

$$y_{\text{Stress}} = \begin{cases} b & \text{baseline intramural stress,} \\ b + p & \text{perturbed from baseline,} \end{cases} \qquad (1)$$

$$y_{\text{Wss}} = \begin{cases} 0.5 & \text{baseline wall shear stress,} \\ 0.5 + p & \text{perturbed from baseline,} \end{cases} \tag{2}$$

$$y_{\text{AngIIin}} = \begin{cases} 0 & \text{baseline infused AngII,} \\ p & \text{perturbed from baseline,} \end{cases} \tag{3}$$

$$y_{\text{SACs, Integrins}} = \begin{cases} b & \text{baseline cell receptors,} \end{cases} \tag{4}$$

where $b \in [0, 1]$ is a normalized basal input value (to be prescribed) and $p \in [0, 1]$ is the magnitude of a normalized perturbation (also to be prescribed), which is added to the baseline level of Stress, Wss, or AngIIin, one at any time. Note that an exogenous AngII input is only present in certain experimental protocols, and is usually compared to controls with no AngII infusion. We therefore set $y_{\text{AngIIin}} = 0$ at baseline, rather than $b$, to represent our control state of the artery. Exogenous AngII is then considered as a perturbation with respect to this control. Note, however, that this does not mean that there is no AngII signaling at baseline, for AngII is also activated by intramural stress (Fig 1). The baseline value for wall shear stress ($y_{\text{Wss}} = 0.5$) was a model choice due the desire to study increased and decreased wall shear stress equally, based also on examination of the corresponding ET1 and NO steady states. This choice ensured capture of known behaviors (*e.g.* of ET1 increasing below, and NO increasing above, baseline wall shear stress).

Experimental results used to parameterize the model and to ensure qualitative validation are shown in Fig 2A; they were collated from a set of 37 papers that report qualitative outcomes in terms of single inputs. Again, note that these papers are separate from the literature used to construct the network, which were focused on single reactions rather than network level input–output relations. An increase or decrease of an output in response to an increased input is represented by upward and downward pointing arrows respectively. In some cases, conflicting results were found in the literature, in which case both arrows are shown. Cases with no observed changes are shown by horizontal lines, and unknown relationships by empty boxes. The values of $b$, $p$, and default network parameters were selected, as described in Methods and S2 Appendix, as single values that provide the best match to these qualitative relations (here, $b = 0.2$ and $p = 0.3$ in Eqs 1–4). Associated model input–output relations, which correspond directly to the experimental results in Fig 2A, are shown in Fig 2B, where absolute increases and decreases relative to baseline steady states are shown by orange and blue respectively. Check marks and crosses denote agreement and disagreement, respectively, between the model and experiments, excluding cases where opposing outcomes have been reported in different studies. The goodness of this parameterization, and thus validation of the model, is revealed by the overall qualitative agreement with the (non-conflicting) experimental findings in all but two cases, both in response to changes in wall shear stress: the model did not predict the experimentally observed upregulation of TGF$\beta$1 by wall shear stress, whereas it predicted a small decrease in MMP9, which was reported not to change in the literature. Note, for the purposes herein, cardiac output (flow) tends not to change appreciably in many cases of hypertension.

We also considered the consistency of qualitative responses under changes in $b$ and $p$ (S2 Appendix), highlighting differential responses that are sensitive to inputs or levels of perturbation. Examples include predicted changes in MMPs, actomyosin activity, and SMC proliferation in response to prescribed increases in stress or exogenous AngII, and predicted changes

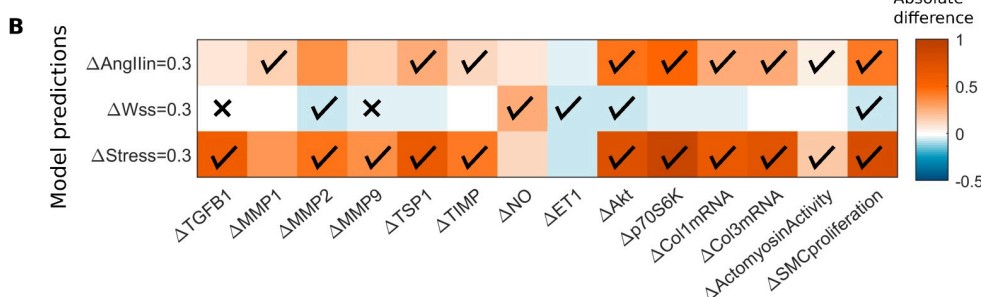

| | | Perturbed Inputs | | | | |
| --- | --- | --- | --- | --- | --- | --- |
| | Stress | Wss | AngIIin | Stress | Wss | AngIIin |
| TGFB1 | ↑ | ↑ | ↑↓ | [24] | [25] | [26, 27] [28] |
| MMP1 | | | ↑ | | | [29] |
| MMP2 | ↑ | ↓— | ↑↓ | [30–32] | [33, 34][35] | [36, 37][38] |
| MMP9 | ↑ | — | ↑↓— | [31, 32] | [35] | [36, 37] [39][40] |
| TSP1 | ↑ | | ↑ | [41] | | [41] |
| TIMP | ↑ | | ↑— | [30, 42] | | [43] [36, 40] |
| NO | | ↑ | | | [44, 45] | |
| ET1 | ↑↓ | ↑↓ | | [46][46] | [47][47] | |
| Akt | ↑ | ↓ | ↑ | [48] | [49] | [50] |
| p70S6K | ↑ | | ↑ | [48] | | [51] |
| Col1mRNA | ↑ | | ↑ | [52, 53] | | [26, 27, 53] |
| Col3mRNA | ↑ | | ↑ | [42, 52] | | [52] |
| ActomyosinActivity | ↑ | | ↑ | [54, 55] | | [56] |
| SMCproliferation | ↑ | ↓ | ↑ | [57] | [25, 49] | [58–60] |

*Experimental Observations*

**Fig 2. Comparison of qualitative experimental input–output relations and model predictions.** A. Experimental input–output relations from the literature ([24–60]) used for qualitative model validation. Increases and decreases of an output in response to each of three inputs are represented by upward and downward arrows respectively. In cases of conflicting results, both are depicted. Cases with no observed changes are shown by horizontal lines, and unknown relationships by empty boxes. For the supporting references, orange indicates upregulation, black indicates no observed change, and blue indicates downregulation. B. Model predicted absolute differences in steady state species activity when the inputs (intramural stress, wall shear stress (Wss), and AngII) are perturbed relative to baseline: orange denotes an increase and blue a decrease relative to the original steady state (white). These simulations correspond directly to the experimental findings, with model parameters tuned to achieve the best qualitative agreement (see S2 Appendix). We denote agreement and disagreement between model and experiments by check marks and crosses, respectively. Default uniform model parameters are $n = 1.25$, $EC_{50} = 0.55$, $b = 0.2$ and $p = 0.3$ (Eqs 1–4), with weights $w = 1$ (see Methods). TGFB: transforming growth factor-$\beta$, MMP: matrix metalloproteinase, TSP1: thrombospondin-1, TIMP: tissue inhibitor of MMPs, NO: nitric oxide, and ET1: endothelin-1.

in TGF$\beta$1 in response to prescribed exogenous AngII. For a subset of these cases, we illustrate this dependence by plotting fold changes in steady state behavior relative to the baseline ($p = 0$) case, as the parameters $b$ and $p$ vary, where $p$ is a stress perturbation (S3 Appendix). We find cases where increases (TGF$\beta$1, TSP1, NO) and decreases (ET1) are found consistently or where both increases and decreases (MMP1, MMP2) can be seen, which can be understood when plotting the behavior in absolute (rather than fold-change) terms (S3 Appendix). We found that the species exhibiting inconsistent qualitative responses exhibited non-monotonic behavior as $b$ and $p$ increase. Qualitative conclusions thus depend on input level, that is, the baseline or point of reference for the comparison, and the magnitude of the perturbation. This

is also illustrated in a simple example (S4 Appendix) in which a non-monotonic input–output relation between TGF$\beta$1 and MMPs occurred due to TIMP inhibition.

## Sensitivity analysis

The present rule-based modeling approach allows considerable control over each species and reaction; interactions can easily be added, removed, or altered (by adding new edges or adjusting the weight parameters) and species can be fully or partially removed via a scaling parameter $Y_{max} \in [0, 1]$ (see Methods). This flexibility is extremely useful for understanding the importance of particular nodes and pathways, and we can simulate downstream effects that may result from abnormal signaling, mutations, or therapeutic interventions. Consider, therefore, a partial knockdown of interior nodes in the network (Fig 3). For each node in turn, we calculate the absolute difference in steady state activity of each species as the value of $Y_{max}$ for that node is reduced from 1 to 0.1. Default parameters ($w = 1$, $n = 1.25$, $EC_{50} = 0.55$) and the uniform initial conditions, $y_0 = 0.2$ (the basal condition), were used for four of the inputs: Stress, AngIIin, SACs, and Integrins, whereas Wss = 0.5 was used for the wall shear stress. Such simulations can be studied further to see direct and indirect consequences when removing specific species, which could occur due to mutations, dysfunction or targeted interventions, such as treatment with losartan, an $AT_1$ receptor blocker. From Fig 3, we see that a knockdown of the $AT_1$ receptor (AT1R, marked by (1)) results in reductions in SMC proliferation and contractile proteins (RhoA, ROCK, MLCK), consistent with experimental studies showing reduced SMC proliferation and contraction [61]. In contrast, an $AT_2$ receptor knockdown (AT2R, marked by (2)) results in upregulation of contractile proteins and SMC proliferation, consistent with experimental observations of its negative regulation of RhoA and ROCK [62], and the opposing effects of this receptor on $AT_1$ receptor signaling [63]. Note here the apparent lack of changes in FAK, Cdc42, Arp2/3 and ActomyosinActivity, and their downstream nodes. These species had low basal values, and their absolute values were modulated only slightly when enforcing $Y_{max} = 0.1$, to an extent not visible on this scale. We show this subset of results in S1 Fig using a different scale.

We also examine downstream effects of reducing ET1 and NO in the model. In an experimental study by Rizvi *et al.* [64], the vasoconstrictor ET1 was observed to stimulate SMC proliferation and collagen type I synthesis but not collagen type III synthesis. Knockdown of ET1 in the present model reduced SMC proliferation and (to a small extent) collagen type I mRNA expression, but not collagen type III mRNA expression. In the experiments, an $ET_A$ receptor antagonist reduced collagen type I synthesis; in the model, knockdown of the ETAR node (marked by (3)) similarly led to a slight reduction in collagen type I mRNA levels. Note that the model shows a decrease in collagen at the mRNA level but not in the total protein due to the concurrent decrease in MMPs and the way in which their interaction with collagen is modeled. In the current formulation, MMPs directly inhibit the synthesis of functional collagen via an 'AND NOT' operation (S1 Appendix) rather than more accurately degrading the protein after its production; the net outcome is the same. More realistic turnover could be accounted for by including additional species that distinguish between newly synthesized collagen and degraded collagen, but was not considered here. An additional consideration is the weight parameter associated with this degradation, which should be adjusted according to experimental data though not done here. Rizvi *et al.* also investigated the effect of the vasodilator NO [60], finding that it inhibited SMC proliferation and collagen type I synthesis, but not collagen type III synthesis. Consistent with these observations, an NO knockdown (*i.e.* the reverse scenario) in the model (marked by (4)) led to increased SMC proliferation and slight increases in collagen type I and III mRNA. The effect is larger in collagen type I mRNA than in collagen

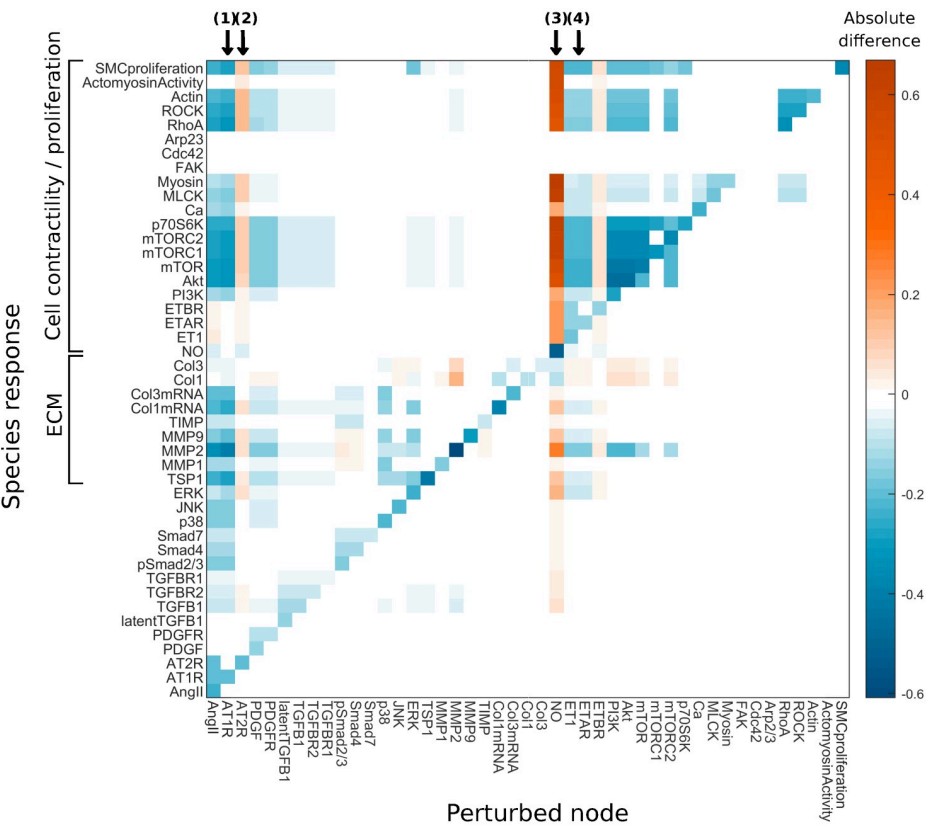

**Fig 3. Network sensitivity analysis as each node is perturbed.** For a separate individual partial knockdown of each of the 45 interior nodes ($Y_{max} = 1$ to $Y_{max} = 0.1$), we calculate absolute differences (knockdown–reference) in steady state activity of every other species ($y$–axis). The marked cases (1)–(4) are discussed in the main text. In both the reference and knockdown cases, uniform initial conditions, $y_0 = 0.2$, are used for four of the inputs: Stress, AngIIin, SACs, and Integrins, whereas Wss = 0.5.

type III mRNA. We further observe that ERK signaling increased, and from the network diagram (Fig 1), note that ERK activates Col1mRNA but not Col3mRNA.

## Case studies

To mimic the experimental study of Ruddy *et al*. [39], we consider different combinations of intramural stress (Eq 1), with values $b$ (low), $b + p$ (intermediate) and $b + 2p$ (high) respectively, and exogenous AngII (Eq 3), with values $b$ (low) and $b + 2p$ (high) respectively. Fig 4 shows outputs for different values of $b \in [0, 1]$ and $p \in [0, 1]$. First, consider a low baseline (left column): $b = 0.1$ and $p = 0.1$. In this case, AngII increases MMP activity for each level of stress. In contrast, for intermediate baselines (middle column, $b = 0.2$, $p = 0.2$), MMP activity decreases in the intermediate and high stress cases (shown by increasing, then decreasing arrows). Finally, for higher baselines (right column, $b = 0.4$, $p = 0.2$), MMP activity decreases for all three stress states (strictly decreasing arrows). These representative parameter choices show the different possible qualitative outcomes, found in all three types of MMPs considered, the second of which resembles observations in [39] in which MT1-MMP and MMP9 promoter activity increase under low tension but decrease under high baseline tension with the addition of AngII (Figs 1 and 3 in [39]). In their study, MMP2 promoter activity (Fig 2 in [39]) did not show this conflicting behavior, but instead showed increased activity more akin to the low

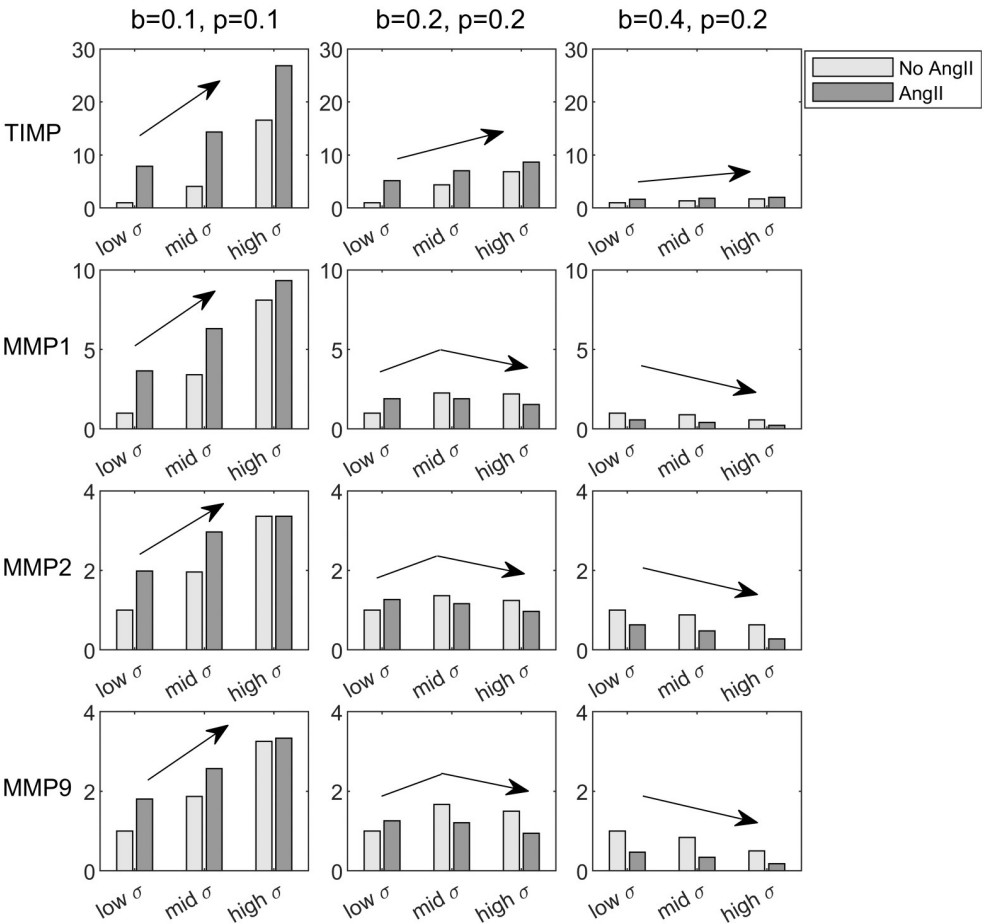

**Fig 4. Species responses to exogenous AngII under three levels of baseline stress.** We show model outputs (relative to the baseline case, Stress = AngII = $b$) for four species of interest in response to three levels of stress, $\sigma$: low ($b$), intermediate ($b + p$), and high ($b + 2p$), as well as low ($b$) and high ($b + 2p$) AngII inputs. Arrows show general trends, with the MMPs exhibiting three different qualitative behaviors, the first two of which were similar to those observed in Figs 1–3 in [39].

baseline case (as in the left hand panels of Fig 4 for MMPs, with strictly increasing arrows). Recall, however, that the model parameters were not tuned to fit these particular experimental data or these differences, which could arise (for example) from different activation weights. It is promising that we are able to observe both of these qualitative outcomes, and future quantitative studies may help to capture these differences. Note, too, that not all species show dose dependent behavior; TIMP shows only increasing or saturating behavior, meaning that it is not as sensitive to baseline conditions. Strictly increasing behavior is also found for TSP1 (S2 Fig), and slight decreases, and therefore conflicts, can be seen in TGF$\beta$1 with the addition of AngII at high baseline stresses (S2 Fig), but this effect is not as apparent as in the MMPs (Fig 4). This result is likely related to the decrease in MMPs, since MMP2 and MMP9 cleave latent TGF$\beta$, and thereby regulate the active form.

These different responses can be characterized fully by plotting steady state outputs as a function of the two inputs, stress and AngII (Fig 5A); we present these as dose-response surfaces. Similar to the cases with one input (S3 Appendix), non-monotonic behavior is seen for all types of MMPs. Namely, as the baseline levels of stress increase, decreasing MMP activity relative to baseline can be expected after further perturbations, which we show by considering

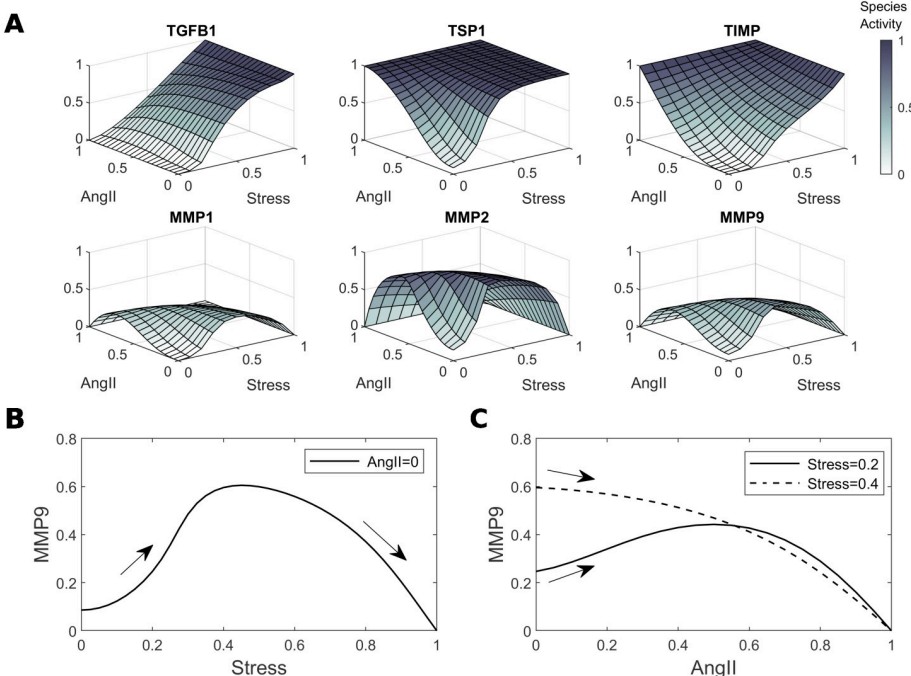

**Fig 5. Model dose-response surfaces to Stress and AngII and consequences of non-monotonicity.** A. Output steady states of 6 species of interest as a function of Stress and AngII inputs, yielding dose-response surfaces. B. Cross-section of the MMP9 surface showing how MMP activity can first increase and then decrease in response to Stress, here with AngII = 0. C. Cross-section of the MMP9 surface showing that intermediate and high baseline Stress can lead to either initial increases or decreases in MMP activity as AngII is applied.

the cross-section of the MMP9 surface at AngII = 0 (Fig 5B). Similarly, AngII perturbations can lead to either increased or decreased MMP activity, the latter more prevalent at higher levels of stress, although this also depends on the magnitude of the perturbation, as shown by considering cross-sections for an intermediate and high baseline stress (Fig 5C). The experimental study of Ruddy *et al.* [39] can be thought of as sampling from these dose-response surfaces, and this experimental design is therefore more useful than single-dose studies for understanding possible non-monotonic input–output responses, though more combinations could be helpful. When moving to quantitative studies, this information could be used to help identify the reference point on the surface, which is currently unknown, and to identify key parameter values, which will affect the shape.

To mimic the experimental study of Wu *et al.* [52], which focuses on matrix production by murine aortic fibroblasts, we examine collagen mRNA levels in the model under three different levels of stress ($y_{\text{Stress}} = \{0.2, 0.3, 0.4\}$) with and without Ang II ($y_{\text{AngIIin}} = \{0, 0.2\}$) (Fig 6). In qualitative agreement with data in [52] (shown here by filled circles, when numerical values were available), stress increases mRNA expression of collagen types I and III in a dose-dependent manner, which is increased further by exogenous AngII. Finally, we simulate a knockdown of p38 MAPK (to 10% maximal activity, $Y_{max} = 0.1$), to mimic the use of the inhibitor SB203580 (Fig 5D,E in [52]). This knockdown attenuates the increased expression of mRNA for Col1 and Col3 induced by stress. In addition to fold changes, we show the corresponding time-courses in S3 Fig. Interestingly, although the fold-change response to AngII was larger in collagen type III mRNA, the absolute differences are similar. In the model, this occurs simply due to a lower basal value of collagen type III mRNA. The response to the p38MAPK

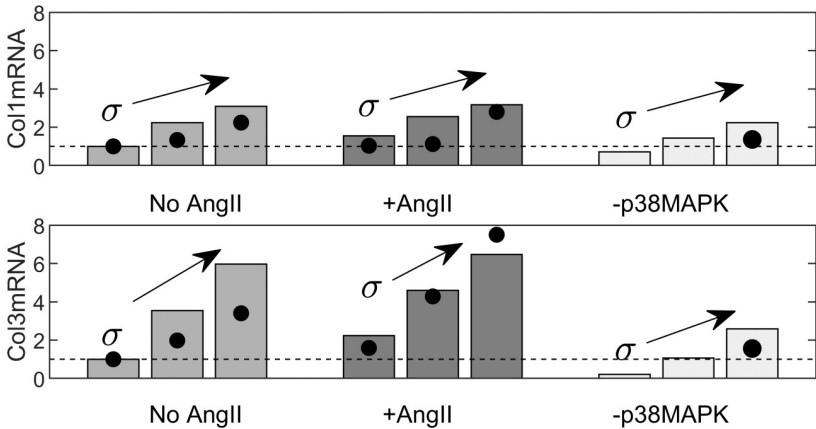

**Fig 6. Model-predicted collagen mRNA expression for control, AngII, and p38 MAPK knockdowns as wall stress increases.** We show fold-change expressions of collagen type I and collagen type III mRNA with three levels of stress ($y_{Stress} = \sigma = \{0.2, 0.3, 0.4\}$), with and without AngII ($y_{AngIIin} = \{0, 0.2\}$). Bars are model outputs, and filled circles correspond to data from [52]. In the absence of AngII, we also simulate a p38 MAPK inhibitor (via a knockdown to 10% maximal activity), which corresponds to findings in Fig 5D,E in [52]. Note that the default Hill parameters were not refined to achieve quantitative agreement, which was considerable nonetheless.

knockdown is, however, more pronounced (in both relative and absolute terms) in collagen type III mRNA, as observed in the experiment (Fig 5D,E in [52]).

## Discussion

Many different models focusing on different conditions have been proposed to study arterial growth and remodeling [65–70]. Among others, we have found phenomenological models to be useful in generating and testing diverse hypotheses fundamental to arterial adaptations [5, 22], in studying arterial disease progression [71, 72], and in the design of tissue engineered constructs and their clinical usage [73, 74]. Nevertheless, tissue-level manifestations arise from molecular and cellular level changes [75–78]. There is, therefore, a pressing need for models that enable one to examine changes in cell phenotype and ECM turnover in terms of cell signaling pathways. As noted above, both kinetic and logic-based models offer considerable promise in this regard. Some models coupling tissue mechanics to cell signaling have been developed using kinetic formulations [4, 79], and provide illustrative examples through parameter studies. Biochemical species—primarily growth factors and proteases—were modeled using either a system of ODEs [4, 80] or reaction–diffusion PDEs [79]. Yet, with time-course data for these species lacking, parameterization and quantitative verification remains a challenge, particularly if more detailed signaling is to be considered in the future.

Here, we implemented a logic-based model building on prior successes in modeling cardiac remodeling [10, 81–83]. This is, to our knowledge, the first such implementation for arterial signaling, with a focus on processes relevant to hypertensive growth and remodeling. Although there is a pressing need for consistent data for particular situations of interest—for example, angiotensin-induced hypertensive remodeling of the murine aorta—we were able to identify 72 papers in the literature that allowed construction of a general network topology (model) and 37 additional papers that allowed the parameter values to be tuned, when assumed to be uniform (*i.e.* taking identical values for all reactions). We then focused on arterial responses to three stimuli (inputs): changes in intramural stress, changes in wall shear stress, and changes in AngII stimulation. Overall, model predictions were qualitatively consistent with findings reported in 83% of the cases (and up to 92% of cases when considering only papers with

consistent findings), similar to that reported for cardiac models [81, 83]. This level of qualitative agreement allows the model to be used with some confidence to investigate effects of various knockdowns of particular nodes or perturbations in various inputs, namely stress and exogenous AngII.

We then focused on two papers of particular relevance to our goal—the paper by Wu *et al.* [52] that investigated intramural cell (adventitial fibroblast) level responses to cyclic stretching with and without AngII stimulation and the paper by Ruddy *et al.* [39] that investigated tissue (whole wall, homogenized transmurally) level responses to different applied loads (in a tension ring test) and AngII simultaneously. In both cases, our model predictions were largely consistent with the experimental findings despite our not attempting to refine the parameter values for these specific studies. Specifically, for different combined doses of intramural stress and AngII, the model predicted, in agreement with [52], that stress increased mRNA expression of collagen types I and III. As in the experimental findings, this effect was dose-dependent, further increased by the addition of AngII, and mediated, in part, by p38 MAPK (Fig 6). Also consistent with the experimental data, the effect of a p38 MAPK knockdown was more pronounced on collagen type III. As studied in [39], we also considered MMP activity for different levels of the stress and AngII inputs, and found different possible qualitative results depending on the baseline level of stress (Fig 4). These model findings appear to result from non-monotonicity, resulting from inhibition reactions, which was demonstrated by analyzing the equations for a simpler illustrative example (S4 Appendix). Similar results were seen in [39] for MT1-MMP and MMP9 promoter activity under low and high baseline tension, but not for MMP2. In addition to this example, non-monotonic input–output relations could explain conflicting experimental findings in the literature, which often only consider single doses and baselines. We emphasize that further data from studies considering consistent baseline conditions, multiple doses, and multiple combined inputs will be essential to characterizing system behaviors further, and for identifying dose-response surfaces, as in Fig 5. These studies would also be essential for determining relative contributions of the different inputs on shared pathways, thereby allowing improved parameterization of the model. In addition to these considerations, future collection of time-course data, which is currently lacking, will prove useful. The model includes a decay timescale, $\tau$, for each species (see S4 Appendix or [84]), which was not adjusted here (similarly to previous works [82–85]) due to lack of temporal data. This is a significant simplification and, whilst order of magnitude estimates have been considered [81], time-course data will help in these parameterization efforts.

Inherent to logic-based approaches is the normalization of species activity to the range [0, 1], meaning that conclusions focus on qualitative rather than quantitative trends. This is often the level of detail that is available from the many different biological assays available, which are based on fold-change responses. Whilst kinetic models may eventually become appropriate as detailed quantitative data become available, tuning to quantitative results can be achieved with logic-based models by adjusting reaction weights and Hill parameters, as demonstrated in the initial formulation for cardiac signaling [84], where comparisons were made to an existing kinetic model. A considerable challenge, however, will be parameter identifiability, due to the numerous pathways with similar functional forms; comprehensive datasets will be needed before such parameter fits can reliably take place. Nevertheless, there is still much insight to be gained from qualitative simulations, even under the simplifying assumptions made here. The flexibility to refine and test different network structures is extremely useful, and qualitative observations can guide and motivate experimental studies. Of particular note are the contradictory experimental results in Fig 2A. Although many possible factors could give rise to these contradictions, our qualitative prediction of non-monotonic responses to input perturbations provides one potential explanation; fold-changes become sensitive to

baseline conditions and perturbation magnitudes (S3 Appendix). Conflicting qualitative results were also seen within a single study by Ruddy *et al.* [39], when AngII was perturbed under varying levels of baseline tension, also consistent with our view on non-monotonicity and the importance of baseline conditions (Figs 4 and 5). This possibility can be tested further via studies with controlled baselines and multiple perturbation magnitudes; more generally, though, this highlights the need for more of these types of studies if fold-change data are to be reliably understood.

In this first generation model, intramural cells are considered together, representing homogenized or bulk responses. This choice is motivated by three observations: first, a long history of modeling the mechanics of elastic arteries via radial homogenizations, whereby mean wall mechanics are captured well using single-layered models [21]; second, by mechano-biological models for arterial G&R [22, 23], where arterial adaptations and maladaptations are captured well using phenomenological models of combined intramural cell mechanosensing; and third, the overwhelming availability of bulk biological data, including qPCR, western blotting, and bulk RNAseq that come from homogenates from the arterial wall, particularly in murine studies wherein it is difficult to separate the three structural layers of the wall. Therefore, whilst still a first order approximation, our model is in a convenient form to be coupled to data-informed G&R models. In future refinements, however, more detailed signaling networks should be considered for each of the three key cell types within the arterial wall: endothelial cells, smooth muscle cells, and fibroblasts, which reside in the intima, media, and adventitia, respectively. The importance of paracrine signaling can then be studied (requiring also a wider collection of co-culture data), and this could then be coupled to multi-layered models of arterial G&R [5]. Indeed, with increasingly more detailed data, there will be a need for further delineation of cellular contributions to wall growth and remodeling; we know, for example, that macrophages and T-cells contribute to fibrotic remodeling in hypertension [86, 87] and both resident and bone marrow derived progenitor cells often contribute, the latter including fibrocytes [87]. Similarly, there are many other species of ECM not considered here: glycoproteins, GAGs, additional MMP subtypes, and ADAMTS, for example, that can and should be added in future models. Of course, the more complex the model, the more difficult its validation, and extensions should take the available data into account.

Arterial remodeling is inherently multiscale, with feedback between cell-level signaling events and slower tissue-level responses, which leads to changes in mechanical stresses over time. In order to capture this feedback, an important future extension is the coupling of network models to tissue-level mechanical descriptions of the arterial wall. Of note, the framework used here is compatible with previous constrained mixture models for G&R [21–23], where the normalized outputs from our network model can directly inform, via appropriate scalings, constituent mass production and removal functions and altered contractility, which were previously modeled using phenomenological functions of intramural and wall shear stress. Similarly, intramural and wall shear stresses calculated from such a G&R formulation will inform the changing inputs to the network. It is critical that this feedback between network outputs and network inputs comes from such a tissue-level model, which includes mass and momentum balance, rather than being included directly in the logic-based model, as only then we can understand the role of material parameters, composition, and other key determinants such as collagen fiber alignment in the modulation of the stresses. Development of a coupled, multiscale, framework will significantly extend the scope of the current model, allowing us to simulate long-term consequences of disrupted signaling, which cannot be captured in a signaling model alone. Similarly, the effect of sustained changes in inputs, arising in hypertension, for example, can then be considered.

In summary, we developed a model that predicts changes in cell phenotype and ECM turnover in response to prescribed changes in three fundamental inputs in arterial mechanobiology, namely tissue-scale intramural stress, wall shear stress, and exogenous AngII. These inputs are particularly important in cases of induced hypertensive aortic remodeling and our preliminary studies show good consistency with available data. Whereas we considered some of the key cells, ECM constituents, and signaling pathways (Smad, MAPK, mTOR, PI3K/Akt, and Rho/ROCK), there is a need to consider additional cellular contributors, matrix constituents, and pathways as well. Most importantly, there is a need to couple the current cell-signaling based model with tissue-level models that include equilibrium solutions that define the evolving states of intramural stress as well as hemodynamic models that define the evolving states of wall shear stress. In this way, we can achieve a coupled fluid-solid-growth model having multi-scale capability and feedback consistent with tissue homeostasis (S4 Fig), with the eventual goal of examining how particular mutations or targeted pharmacological interventions can affect the overall wall mechanics and thus (patho)physiological function. Continued collection of data across scales will enable such modeling and should be given highest attention.

## Methods

We use a graph representation to describe signaling events within the arterial wall, where nodes correspond to species of interest and edges depict relationships between them such as activation and inhibition. To implement a graph-based model, we must understand (i) the components involved and (ii) the way they interact (*i.e.* activation vs inhibition). We constructed a network (Fig 1) from an extensive curation of the literature, and formulated the relations as a set of logic statements (S1 Appendix).

### Logic-based governing equations

Starting from a list of logic statements, we use an approach developed by Kraeutler *et al.* [84], which utilizes weighted normalized Hill functions in a system of nonlinear ordinary differential equations (ODEs), details of which are in S1 Appendix. This method, in which a continuous model is built from a discrete set of rules, builds on prior theoretical work [88] and has since been implemented for several other large-scale signaling studies [81–83, 85]. The method extends concepts from Boolean algebra in which each of $N$ species is represented by a discrete activity level, either 'on' (1) or 'off' (0). In the normalized Hill ODE approach, these activity levels can take any real value within the interval [0, 1]. As with traditional logic frameworks, conditional update rules are supported through the use of three basic operators: conjunction, disjunction, and negation ($\wedge$, $\vee$, $\neg$), also known as 'AND', 'OR' and 'NOT' logic gates, which allow key features of signaling networks, such as activation and inhibition involving multiple components, to be simulated.

In the context of cell signaling, the two elementary processes are activation and inhibition. Sigmoidal activation by a single variable, $X \in [0, 1]$, is modeled by a normalized Hill function of the form

$$F(X) = \frac{BX^n}{K^n + X^n}, \tag{5}$$

where $n$ is the Hill coefficient, controlling the steepness of the function (note: Eq 5 approaches a step function as $n \rightarrow \infty$). The constants $B$ and $K$ enforce the constraints

$$F(0) = 0, \qquad F(1) = 1 \quad \text{and} \quad F(EC_{50}) = 0.5, \tag{6}$$

where $EC_{50}$ is the value of $X$ at which a half-maximal activation occurs, namely,

$$B = \frac{EC_{50}{}^n - 1}{2EC_{50}{}^n - 1} \quad \text{and} \quad K = (B - 1)^{1/n}. \tag{7}$$

As typical of logic-based models, inhibition is modeled by negation, that is $1 - F(X)$.

To consider multivariable activation or inhibition, these functions must be extended by using conditional logic. The conditional 'AND', 'OR' and 'AND NOT' operators are defined as

$$\text{(a)} \ X \wedge Y = F(X)F(Y), \tag{8}$$

$$\text{(b)} \ X \vee Y = F(X) + F(Y) - F(X)F(Y), \tag{9}$$

$$\text{(c)} \ X \wedge \neg Y = F(X)(1 - F(Y)). \tag{10}$$

Additionally, these operators can be used recursively to construct more complex regulatory statements, involving more than two components.

Reaction weights can be introduced into the normalized Hill ODE formulation to better fit quantitative experimental data; the governing equations become more similar (though different in underlying assumptions) to kinetic models, since edge weights can be tuned as more information becomes known about individual reactions [84]. Weighted reactions are also useful for exploring different network topologies: edges can be modeled as defective or removed by lowering or setting reaction weights to zero. Additionally, each node has a decay timescale $\tau$ and a maximal activity level, $Y_{max} \in [0, 1]$, as discussed fully in [84]; we also provide a detailed example of model construction, governing equations, and solutions for an illustrative reduced system in S4 Appendix. By default, $Y_{max} = 1$; however, external interventions such as full or partial knockdowns of a species can be simulated by lowering this value.

The precise form of the weighted normalized Hill ODEs depends on the set of rules governing each variable, but are built in a modular fashion using Eq 5 for activation, its negation for inhibition, and the conditional logic operations in Eqs 8–10. In general, each reaction is then scaled by a weight parameter, $w$. This can be seen for our illustrative system in S4 Appendix.

**Default parameters.** In this first implementation of the model, we assumed that the reaction weights and Hill parameters ($w$, $n$, $EC_{50}$) are uniform across the network (*i.e.* they take identical values for all reactions), consistent with demonstrated successes by others using this assumption [81–85]. Additionally, let $w = 1$ for each reaction and $\tau = 1$ for each species. The strength and functional forms for activation and inhibition are therefore identical for each reaction, as in Boolean models, and we associate each species with the same decay timescale. These assumptions were able to capture well the qualitative network behaviors reported in previous experimental studies, suggesting that the basic model structure reflects well the arterial wall under the conditions of interest. These simplifying assumptions will nevertheless need to be adjusted when there is increased access to quantitative data, including time-courses. To tune the model parameters using input–output simulations (Fig 2B), we conducted a parameter sweep to find the baseline values and perturbation magnitudes of the inputs, denoted by $b$ and $p$, respectively (Eqs 1–4), and the Hill parameters ($n$ and $EC_{50}$), that provided the best agreement with experimental input–output relations (Fig 2A). For each parameter set, we quantified the percentage of input–output relations that qualitatively matched between model and experiment and, based on this analysis, shown in more detail in S2 Appendix, we selected $n = 1.25$, $EC_{50} = 0.55$, $b = 0.2$ and $p = 0.3$ as default parameters.

### Implementation

The open-source code 'Netflux' (https://github.com/saucermanlab/Netflux) provides an automated means of converting rule-based descriptions into weighted Hill ODEs of the type described above and formulated in [84]. The code can be implemented and modified in MATLAB. It was used here for two purposes: (i) to generate the initial system of ODEs for the full network (Fig 1) from our list of logic statements (S1 Appendix) and (ii) to generate an .xgmml file which allows network visualization in Cytoscape [19]. Modifications to the code allowed manual control over input variables (as described below), and our codes for simulating, plotting, and analyzing our model system are available at https://github.com/irons-l/arterialsignaling.

## Supporting information

**S1 Fig. Sensitivity analysis for perturbed nodes.** A subset of results from Fig 3, with a different scale for improved visualization of small changes.
(PDF)

**S2 Fig. Additional species' responses to AngII.** TSP1 and TGFB1 responses to exogenous AngII under three levels of baseline stress, corresponding to Fig 4 in the main text.
(PDF)

**S3 Fig. Time-courses of collagen mRNA levels.** An additional figure showing time-courses associated with the steady state model results in Fig 6.
(PDF)

**S4 Fig. Vascular homeostasis.** An illustrative schematic of the bio-chemo-mechanical feedback system for tissue homeostasis.
(PDF)

**S1 Appendix. Logic statements and supporting literature.** Species abbreviations, logic statements, and supporting references used in constructing the network structure shown in Fig 1.
(PDF)

**S2 Appendix. Selection of default parameters.** Description and supporting figures for the process of selecting the optimal default Hill parameters.
(PDF)

**S3 Appendix. Sensitivity of fold-change responses to baseline conditions.** Fold-change responses and absolute activity of several species of interest as baseline and perturbation magnitudes vary. We show that non-monotonic responses to inputs underlie conflicting fold-change responses.
(PDF)

**S4 Appendix. Simple illustrative model.** An illustrative model used to demonstrate the process of formulating logic statements, generating the corresponding system of normalized Hill ODEs, and calculating the system steady states. In this example, we show that inhibition can lead to a non-monotonic input–output relation, and we illustrate how conflicting fold-change measurements can result.
(PDF)

**S5 Appendix. Sensitivity to Hill parameters.** We demonstrate the role of Hill parameters in signal propagation, focusing on a linear cascade. We show how the choice of $EC_{50}$ can either

lead to decay, amplification, or preservation of signal strength.
(PDF)

## Author Contributions

**Conceptualization:** Linda Irons, Jay D. Humphrey.

**Data curation:** Linda Irons.

**Formal analysis:** Linda Irons.

**Funding acquisition:** Jay D. Humphrey.

**Validation:** Linda Irons.

**Visualization:** Linda Irons.

**Writing – original draft:** Linda Irons, Jay D. Humphrey.

**Writing – review & editing:** Linda Irons, Jay D. Humphrey.

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
