## [Decision Letter · Decision Letter 0]

23 Mar 2020

Dear Dr Irons,

Thank you very much for submitting your manuscript "Cell signaling model for arterial mechanobiology" for consideration at PLOS Computational Biology.

As with all papers reviewed by the journal, your manuscript was reviewed by members of the editorial board and by several independent reviewers.The editors and reviewers noted positive contributions of this model to the field, but several important concerns were raised including the significance of the biological insights as presented, which is an important criterion for this journal. In light of the reviews (below this email), we would like to invite the resubmission of a significantly-revised version that takes into account the reviewers' comments. 

We cannot make any decision about publication until we have seen the revised manuscript and your response to the reviewers' comments. Your revised manuscript is also likely to be sent to reviewers for further evaluation.

Sincerely,

Jeffrey J. Saucerman

Associate Editor

PLOS Computational Biology

Feilim Mac Gabhann

Editor-in-Chief

PLOS Computational Biology

Reviewer's Responses to Questions

**Comments to the Authors:**

Reviewer #1: The authors present an excellent study in which they create a logic-driven computational model of the cell signaling network for growth and remodeling (G&R) of arteries. The authors based their network topology on a comprehensive literature review. Adjusting their parameters, they are able to validate their model by comparing predictions against different reports in the literature. The advantage of using a logic-driven approach is that the authors can validate their model based on qualitative behavior (e.g. a given output increases, stays constant, or decreases), which allows them to compare their predictions with a wide set of experiments from the literature. The authors then explore the main features of their network and perform a well thought out sensitivity analysis. Overall the article is excellent.

I do have one major comment that requires more discussion. The authors present their cell signaling network, but are not specific about the cell population they model. At two points in the manuscript it seems that the regulatory network is specific for Smooth Muscle Cells (SMC), but at some other instances it is clear that they refer to endothelial cells. Most of the literature cited is indeed related to SMC response to mechanical input, but part of the network corresponds to endothelial cells, particularly nitric oxide (NO) dynamics and wall shear stress (WSS) mechanosensing. The distinction is important, since regulatory networks have cell-type specific components, but the network shown in Figure 1 mixes extra-celullar and intra-cellular signals and for the two different cell types. The article needs a clearer demarcation of the network components.

Reviewer #2: This work presents a novel cell signaling network model to predict arterial smooth muscle cell responses to stimulation by mechanical stresses and angiotensin. Employing a logic-based, ordinary differential equation modeling approach that was previously developed for cardiac cell signaling, this current model predicts changes in signaling pathway activities and cellular outputs including matrix content, cell proliferation, and cell contractility. Excitingly, the model is able to correctly predict (qualitatively at least) a wide variety of input-output combinations reported in previous literature experiments, while also agreeing with several past studies of combined angiotensin + mechanical stimulation at the cell and tissue levels. The paper is very well written with a clear and rational organization, the findings are well supported and interesting, and the modeling approach (while previously used for cardiac cells) is innovative within the arterial mechanobiology field. I have only a few minor comments for further improving this manuscript:

Minor Critiques:

1. Further clarification of smooth muscle cell vs. endothelial cell signaling:

The authors state in lines 45-47 that “… we designed our model to consider simultaneously the potential roles of altered wall stresses, wall shear stresses, and AngII infusion on changes in SMC phenotype…”, and the authors also recognize in lines 61-62 that "wall shear stress is sensed primarily by endothelial cells…” Given these comments (and ‘Remark 1’ in the supplementary material), I presume that WSS-related reactions in the network are meant to capture endothelial cell production of NO & ET1 which then stimulate the rest of the network meant to capture smooth muscle cell signaling. It is unclear, however, how this endothelial-smooth muscle cell interaction affected the model fitting and validation - in other words, were all WSS simulations compared to literature data from experiments that contained both SMCs & ECs (in vivo, ex vivo, in vitro co-culture)? Additional comments regarding these SMC-EC interactions in the main text (and perhaps Figure 1) could improve clarity on this point.

2. Criteria for network reaction rules:

A large number of past literature studies were used to assemble the network model structure, but it is unclear whether any particular criteria were used as thresholds to decide what rules to include/exclude. For example, were all of these studies performed with arterial smooth muscle cells, or were cells of any kind acceptable, etc.?

3. WSS and AngII baseline assumptions:

Equation 2 lists yWss baseline equal to 0.5, and Equation 3 lists yAngIIin baseline equal to zero. It seems unclear why these two baseline values are not set as the parameter b, which is used as the baseline parameter for yStress and other cell receptors. Clarifying the justification and implication for these assumptions could help support this choice.

4. Knockdown simulations:

In Figure 5, a few of the knockdown simulations (e.g., Arp23, Cdc42, FAK) produced no changes in any of the nodes (including the knocked-down nodes themselves). This seems surprising and worthy of further explanation (particularly for FAK, which is connected to a number of downstream nodes).

Reviewer #3: The authors use a network to analyze how arteries grow and remodel in response to factors like wall stress, angiotensin II and pressure. This is to study both healthy and maladaptive responses in vessels using signaling pathways. They have found good qualitative agreement of the model with papers previously published about the system and did a parameter sweep to optimize values for parameters like n and EC50. However, the manuscript does not convey clearly how the model can be used to study the system being described. It is not clear what cells signaling mechanisms of the arterial wall are being considered in this study. As noted below, the figures do not have clear conclusions one can draw from them, nor is it clear how model validation was conducted. In addition, the paper makes biological conclusions without acknowledging limitations of the model, and does not adequately describe how this cell signaling network can represent a vessel undergoing growth and remodeling.

Major Comments:

It is ambiguous what system this cell signaling model applies to. Is the model supposed to be representing all of the vessel wall? Do SMCs have a role in ECM deposition? Is this a model taking into account multiple cell types? It is hard to see how Figure 1 can be applied to the entire vessel wall when there are cell receptors in the model.

To better illustrate the model prediction capabilities, Figure 2 would have to include the direct comparison to model results. Addressing this will help emphasize the capabilities of the model, which remain unclear.

Model validation details are missing in the manuscript. It is unclear whether the set of 37 papers were used for construction or only for validation. This is an important component of the model that should be included in the methods

It is unclear what the importance of Figure 3 is. The steady state behavior described in the figure does not add much on their own and the 3D graphs are confusing to interpret. Unless the authors find the need to further describe the contribution of this state to the G&R modeling efforts, it is not clear that this figure should be included.

Figures 6 and 8 could be annotated to show how these model prediction results compare to qualitative experimental data.

In the Discussion Section, it is not clear what biological conclusions can be made based on this model since the Saucerman model is normalized from 0 to 1,

Is there enough data in the literature to make strong conclusions about what the model agrees with and does not? There are contradictory experiments, what does that mean for the confidence in the model?

In general, there is not much discussion of the limitations of this type of modeling to study this system

It is not clear how the species were chosen, what is the scope of the G&R model, and what can this model be applied to. Addressing these questions will significantly improve the manuscript.

To add model significance, the work could benefit from more inhibition studies with time courses to compare against

The authors propose to use this cell signaling model for growth and remodeling. However, it is not clear where the feedback in the signaling model comes into play. How are the authors accounting for the constant adapting and remodeling of the vessels?

Minor Comments:

Figure 1 could be made more clear by differentiating cell signaling molecules from receptors vs transcription factors vs mRNA, etc.

MMP9 is also involved in ECM remodeling, thus is there a reason MMP9 does not have the same effect as the other MMPs?

Figure 5 is missing labels: the color bar is not labelled, the y-axis is not labelled, and “knockdowns” is perhaps not a great descriptor of the x-axis since these are specific nodes that are knocked down.

The Figure 7 color bar is missing its label.

Section 2.3 could better be described as a sensitivity analysis rather than “knockdowns”

Was there an attempt to vary the parameter of tau? If not, it should be justified and described as a limitation.

**Have all data underlying the figures and results presented in the manuscript been provided?**

Reviewer #1: Yes

Reviewer #2: Yes

Reviewer #3: Yes

PLOS authors have the option to publish the peer review history of their article (what does this mean?). If published, this will include your full peer review and any attached files.

Reviewer #1: Yes: Adrian Buganza Tepole

Reviewer #2: Yes: William Richardson

Reviewer #3: No
---

## [Decision Letter · Decision Letter 1]

28 May 2020

Dear Dr Irons,

Thank you very much for submitting your manuscript "Cell signaling model for arterial mechanobiology" for consideration at PLOS Computational Biology. As with all papers reviewed by the journal, your manuscript was reviewed by members of the editorial board and by several independent reviewers. The reviewers appreciated the attention to an important topic. Based on the reviews, we are likely to accept this manuscript for publication, providing that you modify the manuscript according to the review recommendations.

Based on our editorial review, the new concerns of Reviewer #3 are related to items that were at least partially addressed in Revision 1. However, the manuscript would benefit from some additional clarification, particularly related to the model validation papers and how they were used to further refine the default parameters for subsequent figures.

Sincerely,

Jeffrey J. Saucerman

Associate Editor

PLOS Computational Biology

Feilim Mac Gabhann

Editor-in-Chief

PLOS Computational Biology

[LINK]

Reviewer's Responses to Questions

**Comments to the Authors:**

Reviewer #1: I would like to commend the authors for this nice work. My main concern in the initial version of the paper was the lack of specification about the different cell types in the regulatory network model. In my experience, regulatory networks are cell-specific and I raised that concern before. However, the authors argue that models of growth and remodeling (G&R) of the artery homogenize the different layers into a single set of phenomenological equations. Therefore, a homogenized regulatory network would indeed be a natural way to couple to the author's G&R framework to this, more mechanistic model. While I think there is still room for further development with more detail on the regulatory network, dividing into the different cell types, I understand the authors' position and recognize that the revised version has several clarifications about this homogenization and its limitations. Therefore I am satisfied with the revised version of this manuscript.

Reviewer #2: This revised manuscript presents a novel cell signaling network model to predict arterial smooth muscle cell responses to stimulation by mechanical stresses and angiotensin. The revision is highly responsive to the reviewer comments, addresses all my concerns, and has further improved into an even more excellent study.

Reviewer #3: The authors’ revisions have generally strengthened the manuscript and show better comparison of their model to existing data. Their reworking of Figure 1 and explanation shows a clearer picture of the system they are trying to model. The additions to the discussion help acknowledge the limitations of this type of model.

While the authors have responded to our comments, this reviewer finds that the methods of the model construction and qualitative validation carried out lack important details that need to be addressed.

It would be more clear if authors can indicate what the inclusion of 37 independent papers were used for. As of right now, it is not clear if these papers were used for parameterizing the model after using the 72 studies to construct the model or purely for validation.

The authors describe the parameters as uniform across the network. It would be best if they specified whether the parameters were set to the same initial values and then perturbed independently for each species. Otherwise, it is unclear if the authors assume a uniform distribution of the parameters rather than a single default value.

Given that the authors do not include the system of differential equations and thus do not define tau, they should refer to the original paper where the equations are presented for the reader to reproduce their results.

It would be useful to describe in more detail how the parameter p is perturbed - did the perturbation follow a distribution (i.e, uniform distribution between 0 and 1, Gaussian, etc)?

It is unclear what the criteria are for the use of “best data available” in page 3 line 45. It would be important to include this in the text.

While the reviewers appreciate the authors trying to connect this signaling network model to their tissue-level mechanics, and thus focus on the global, mean, arterial mechanics (homogeneous walls), it is important to clarify that the fibroblasts are primarily adventitial fibroblasts rather than circulating fibroblasts such as it is done in the case studies.

The authors revised figures and their caption and made them more clear in the revised version. However, Figure 3 is not well justified and could be moved to the supplement. Figure 8 is also not well justified to be included in the methods as its results are not unique to the paper.

Minor comments:

Page 17, line 398 ‘lead’ should read ‘leads’

Page 19, line 442 - The authors refer to using a Hill approach but this could be rephrased as it is not completely clear what they mean.

Figure 5: The addition to the caption is worded confusingly and could be improved with more clarity, is it referring to the first two dose-dependent qualitative behaviors or the baseline and first dose-dependent behavior? Also, best if the authors only included a few panels to emphasize important results and moved the rest to the supplement.

Figure 7: Since data from the citation 52 are included in the AngII model predictions, it would be more clear to include the data for -p38MAPK. These data can be found via tools such as WebPlotDigitizer.

**Have all data underlying the figures and results presented in the manuscript been provided?**

Reviewer #1: Yes

Reviewer #2: Yes

Reviewer #3: Yes

PLOS authors have the option to publish the peer review history of their article (what does this mean?). If published, this will include your full peer review and any attached files.

Reviewer #1: No

Reviewer #2: Yes: Will Richardson

Reviewer #3: No
---

## [Editor Report · Decision Letter 2]

13 Jul 2020

Dear Dr Irons,

Thank you very much for submitting your manuscript "Cell signaling model for arterial mechanobiology" for consideration at PLOS Computational Biology. As with all papers reviewed by the journal, your manuscript was reviewed by members of the editorial board and by several independent reviewers. The reviewers appreciated the attention to an important topic. Based on the reviews, we are likely to accept this manuscript for publication, providing that you modify the manuscript according to the review recommendations.

All reviewer critiques have now been adequately addressed, and the authors are to be commended on a strong manuscript. In order for this paper to reach its potential, it is important that the model is fully available for others to test and extend. The data availability statement currently states that "All relevant data are within the manuscript and its Supporting Information files". However, the actual data and code are not included in the Supplement. We would like to see these included either in the supplement, or preferably in a repository. This document may be helpful: 

https://journals.plos.org/plosone/s/materials-and-software-sharing

https://journals.plos.org/plosone/s/data-availability

Sincerely,

Jeffrey J. Saucerman

Associate Editor

PLOS Computational Biology

Feilim Mac Gabhann

Editor-in-Chief

PLOS Computational Biology

[LINK]
---

## [Editor Report · Decision Letter 3]

17 Jul 2020

Dear Dr Irons,

We are pleased to inform you that your manuscript 'Cell signaling model for arterial mechanobiology' has been provisionally accepted for publication in PLOS Computational Biology.

Best regards,

Jeffrey J. Saucerman

Associate Editor

PLOS Computational Biology

Feilim Mac Gabhann

Editor-in-Chief

PLOS Computational Biology

---

## [Editor Report · Acceptance letter]

18 Aug 2020

PCOMPBIOL-D-20-00242R3 

Cell signaling model for arterial mechanobiology

Dear Dr Irons,

I am pleased to inform you that your manuscript has been formally accepted for publication in PLOS Computational Biology. Your manuscript is now with our production department and you will be notified of the publication date in due course.

With kind regards,

Matt Lyles
